# Distinct Inflammatory and Oxidative Effects of Diabetes Mellitus and Hypothyroidism in the Lacrimal Functional Unit

**DOI:** 10.3390/ijms24086974

**Published:** 2023-04-10

**Authors:** Jacqueline Ferreira Faustino-Barros, Ariane Mirela Saranzo Sant’Ana, Lara Cristina Dias, Adriana de Andrade Batista Murashima, Lilian Eslaine Costa Mendes da Silva, Marina Zílio Fantucci, Denny Marcos Garcia, Eduardo Melani Rocha

**Affiliations:** Department of Ophthalmology, Otorhinolaryngology and Head and Neck Surgery, Ribeirão Preto Medical School, University of São Paulo, Ribeirao Preto 14049-900, Brazil

**Keywords:** animal models, diabetes mellitus, dry eye, hypothyroidism, lacrimal functional unit

## Abstract

Diabetes mellitus (DM) and hypothyroidism (HT) are prevalent diseases associated with dry eye (DE). Their impact on the lacrimal functional unit (LFU) is poorly known. This work evaluates the changes in the LFU in DM and HT. Adult male *Wistar* rats had the disease induced as follows: (a) DM: streptozotocin and (b) HT: methimazole. The tear film (TF) and blood osmolarity were measured. Cytokine mRNA was compared in the lacrimal gland (LG), trigeminal ganglion (TG), and cornea (CO). Oxidative enzymes were evaluated in the LG. The DM group showed lower tear secretion (*p* = 0.02) and higher blood osmolarity (*p* < 0.001). The DM group presented lower mRNA expression of *TRPV1* in the cornea (*p* = 0.03), higher *Il1b* mRNA expression (*p* = 0.03), and higher catalase activity in the LG (*p* < 0.001). The DM group presented higher *Il6* mRNA expression in the TG (*p* = 0.02). The HT group showed higher TF osmolarity (*p* < 0.001), lower expression of *Mmp9* mRNA in the CO (*p* < 0.001), higher catalase activity in the LG (*p* = 0.002), and higher expression of *Il1b* mRNA in the TG (*p* = 0.004). The findings revealed that DM and HT induce distinct compromises to the LG and the entire LFU.

## 1. Introduction

Several diseases with distinct pathophysiological mechanisms and epidemiological features lead to dry eye (DE), turning DE into one of the most common health problems worldwide [1,2]. The variability in causes and time length of manifestations may explain why no particular element presents a high predictive value in the diagnosis, and the clinical profile of DE is distinct in different diseases [3,4,5]. It is worthwhile to investigate each disease’s particular characteristics that cause DE, including those associated with a hormone deficiency, to allow better-customized therapies.

The lacrimal functional unit (LFU) system includes the LG, the cornea (CO), the conjunctiva (therefore, the ocular surface), the Meibomian glands, the eyelids, and the sensory nerves. These sensory nerves converge to the trigeminal ganglion (TG) and efferent nerves that bring autonomic inputs, integrating tear secretion into the central nervous system [6]. The LFU keeps the eye surface healthy through tear secretion in terms of volume and content, ensuring protection, lubrication, and renewal of corneal epithelial cells, responding to environmental, neural, and hormonal influences [6]. The damage or disease of any component of the LFU can destabilize the tear film and lead to DE [6].

DM is a common growing condition that causes DE with disease progression [7]. Previous works with streptozotocin-induced DM in rats reveal that these models develop DE around the 8th week of the disease and present markers of oxidative stress, advanced glycated end products, and inflammatory markers in the LG of DM [8]. The tear flow, secretory elements, and LG weight are also reduced in DM [9,10].

HT is common, where hormone deprivation and inflammatory events induce DE [3,6,9,11,12,13]. The mechanisms observed in rat models where HT was induced with methimazole also involve oxidative damage, diminished tear flow, and changes in the CO epithelia by the 5th week of disease induction. Moreover, LG shrinking was reported [14].

In our group’s recent works, we observed the low predictive value of tests for DE diagnosis and their weight in the distinction among different diseases [4,7]. This study hypothesizes that early phases of systemic diseases induced by different hormone deprivation induce DE through distinct mechanisms, affecting the ocular surface, the LG, and the whole LFU. These different triggers lead to differences in the expression of inflammatory and tissue repair mediators and oxidative stress modulators responsible for the different clinical manifestations of DE.

Therefore, this study aims to evaluate the functional and molecular inflammatory aspects of the LG, CO, and TG in the following two rat models of DE: DM and HT.

## 2. Results

### 2.1. Functional and Laboratory In Vivo Findings

Three functional tests were used for the DE functional evaluation of the animal models compared to the control group (CG) at their respective evaluation periods. The eye wipe test measured corneal sensitivity in response to capsaicin. The groups did not show a significant difference from the CG; the DM group showed a tendency of hyperalgesia, and the HT group showed a tendency of hypoalgesia or to be normal (Table 1). The phenol red thread presented lower tear flow in the DM group (*n* = 10/group, *p* = 0.02), (Table 1). Corneal fluorescein staining did not show differences between the DM, HT, and control groups.

The blood and tears osmolarity revealed significant blood hyperosmolarity in the DM group and tear hyperosmolarity in the HT group, with a trend of higher osmolarity in the DM group, but this was not significant (*n* = 10/group; *p* < 0.001 and *p* < 0.001, respectively) (Table 1).

### 2.2. Histological Findings

The histological sections revealed that the LG structure was similar in the three groups (data not shown). However, the corneal epithelium was significantly thicker in the DM group compared to the CG (*p* = 0.003) (*n* = 10/group) (Table 1).

### 2.3. Oxidative Stress and Antioxidant Enzymes

Oxidative stress imposed by the metabolic challenge of DM and HT in the LG was compared using glutathione disulfide (oxidated glutathione) and catalase activity. Catalase, but not glutathione disulfide, was significantly higher in the DM and HT groups compared to the CG (*n* = 10/group; DM: *p* < 0.001 and HT: *p* = 0.002) (Table 1).

### 2.4. qPCR of Pro-Inflammatory and Pro-Mitotic Cytokines

The qPCR in the CO revealed a reduction of *Mmp9* in the HT group (*n* = 10/group, *p* < 0.001) and no changes in *Il1b*, *Il6*, and *Tnfa* in both groups compared to the CG group (Figure 1).

The qPCR in the CO measured the mRNA expression of TRPV1, which is the calcium channel sensitive to capsaicin and other sensory inputs. This showed lower expression in the DM group (*n* = 10/group, *p* = 0.03) and was lower but not significant in the HT group (Figure 1E).

In the LG, the only pro-inflammatory cytokine that increased mRNA levels was *Il1b* in the DM group compared to the CG (*n* = 10/group, *p* = 0.03). The other cytokines did not present a significant difference from the CG (Figure 2).

The pro-mitotic elements with higher mRNA expression in the LG were *Runx3* in the HT group (*n* = 10/group, *p* = 0.01). On the other hand, *Runx1* presented significantly lower expression in the DM group (*n* = 10/group, *p* = 0.02) (Figure 3). The other pro-mitotic elements’ mRNA expression (*Bmp7*, *Fgf10*, and *Smad1*) was similar among the DM, HT, and control groups.

In the TG, among the four cytokines tested, *Il6* mRNA was higher in the DM group, and *Il1b* mRNA was higher in the HT group (*n* = 10/group; *p* = 0.02 and *p* = 0.004, respectively) (Figure 4).

## 3. Discussion

The present work confirmed that DM induced by streptozotocin and HT induced by methimazole are rat models for DE caused by hormone deficiency as a lower tear flow in DM and a higher tear osmolarity in HT were observed [8,9,11]. Moreover, we have shown that both models show distinct manifestations and that DE associated with those diseases involves not just the LG but also the whole LFU, including pro-inflammatory cytokine mRNA rise in the TG, oxidative stress enzymes, and pro-mitotic mediators mRNA changes in the LG.

The DM model presented lower tear flow and high blood osmolarity, and a trend in increasing the mean tear film osmolarity was also observed in clinical reports of DE in DM [3,9,15,16,17,18].

One interesting point was the trend in hyperesthesia observed using the eye wipe test in the DM group, which was associated with lower levels of the *TRPV1* receptors sensitive to CAP in the cornea. The complex mechanism of hyperesthesia in DM may involve higher levels of *Il6* in the TG of the DM group. These observations indicate a mechanism of neuropathic pain in DM involving inflammation of the TG and the central nervous system, which may reflect corneal neuropathy and thicker corneal epithelia [14,19,20,21,22].

The oxidative damage of the LG observed in the DM group was revealed by higher levels of catalase activity in this gland as a response to high oxidative stress in various tissues and organs, including exocrine glands, observed in previous clinical and experimental works in DM [8,9,15,23,24,25]. Oxidative damage may take part in the compromising function, inflammation, and, therefore, in the mechanisms of DE. These observations would prove helpful in addressing the therapeutic strategy for DE in DM.

The HT group presented a trend of hypoesthesia and higher TF osmolarity in the early phase of the disease at 5 weeks of HT. The tear flow showed large variability in the group, confirming the low predictive value of the PRT test [26]. The functional manifestations of DE in the HT and DM groups revealed distinct clinical profiles, confirming the large spectrum of DE manifestations associated with different human diseases and the challenge of a gold-standard diagnostic tool [3,27].

The significantly lower levels of MMP9 mRNA in the corneas of the HT group and the lack of changes of other cytokines compared to the control group suggest that inflammation is not a critical early event in the cornea tissue in DE. Interestingly, inflammation is relevant in the TG of the same group, as discussed below. The expression of the *TRPV1* receptor in the CO tissue was not significantly different from the CG, which disagrees with a recent in vitro study, in which short-term hyperosmolarity induced higher levels of pro-inflammatory cytokines, suggesting compensatory mechanisms in vivo [28]. A trend but not significant change in the pro-inflammatory cytokines in the LG, different from DM, suggests that inflammation of this exocrine gland is not a key factor in the early phase of the disease in the LG in response to HT. However, the higher level of pro-inflammatory cytokine mRNA *Il1b* in the TG indicates the impact of HT in the whole LFU, including the neural pathways. Moreover, the trend of CO hypoesthesia in the eye wipe test induced by capsaicin in the HT group, as a consequence of thyroid hormone deprivation, may be explained by decreased peripheral nerve conduction in HT, as recently observed in humans [29]. Further studies in advanced phases of the disease will be able to clarify the details of this association.

Oxidative stress, as part of the mechanism of the disease in the LG, was confirmed in both groups by higher catalase activity compared to the CG [11,30].

In the HT group, the expression of mRNA of regenerative or pro-mitotic markers tested in the LG, *Runx3* was significantly higher than in the CG. In the DM group, the mRNA of *Runx1* was lower in the LG. It may indicate an attempt to induce recovery of the LG tissue weight, structure, and functional activity in the early phase of both diseases, which seems inefficient as DE persists with the progression of those chronic diseases when untreated [11,31,32,33].

This study’s weaknesses are the lack of long-term DE and the response to DM and HT therapy. Moreover, considering the higher prevalence of DE in the female sex in several species, the present analysis with male rats is limited in identifying the mechanisms of DE in both sexes.

In summary, the present work identified elements of the inflammatory, anti-oxidant, and neural pathways on the mechanisms and the targets of the effects of insulin and thyroid hormone deprivation on the LFU, including functional and organ changes. We also observed that several aspects of DE are distinct among DM and HT diseases and variable inside the groups studied, despite the intended homogeneity achieved by the experimental animal assays. It was not our primary intention to confront HT versus DM in terms of the severity of manifestations; that is why all comparisons were made in paired analysis with the CG, and both conditions were analyzed at different time points (i.e., 5 weeks for HT and 8 weeks for DM) [9,11]. Although some parallels can be inferred, the study design limits direct comparisons.

The present data show no difference in the eye wipe test between the DM, HT, and control groups, a high standard deviation on tear flow values (phenol red thread) in the HT group, and the tear film osmolarity results in the DM group. They reveal that clinical or functional tests are limited in characterizing DE once applied as a single test. This variability of results in distinct groups, even in experimental studies where we can homogenize the genetic background, the time course to measure the parameters in the initial phase of the DE disease, and the environmental conditions, is challenging in studies addressing DE mechanisms [3,9,26,27,34].

## 4. Material and Methods

### 4.1. Animals and Study Design

In all groups, we used *Wistar* male rats, at 7–8 weeks of life, weighing 220–250 g. They were obtained from the Animal Breeding Center of the Ribeirao Preto Medical School, University of Sao Paulo (Ribeirao Preto, Sao Paulo, Brazil). The animal experiments were approved by the Committee for Animal Use of the Ribeirao Preto Medical School, University of Sao Paulo (Protocol 018/2012).

Diabetes mellitus (DM) (*n* = 10) was induced using a single intravenous injection in the tail vein of 60 mg/kg *streptozotocin* (Sigma, St. Louis, MO, USA) in 0.01 M sodium citrate buffer to achieve the final concentration after a 12 h fast. The time between induction and the experimental procedure and tissue harvesting was eight weeks to achieve the early phase of DE in the model [15].

Hypothyroidism (HT) (*n* = 10): The animals received *methimazole* (Biolab Sanus farmacêutica Ltda., Taboão da Serra, SP, Brazil) in drinking water at 500 mg/l continuously for five weeks. The experimental period was chosen to obtain evidence of the early phase of DE in HT [11]. 

Animals in the CG received an injection of citrate buffer in the tail vein and tap water.

The rats were housed in cages at controlled temperatures (23 °C ± 2 °C) on a light–dark cycle of 12 h. Animals had access to standard rodent chow ad libitum. All water sources for the HT group contained the anti-thyroid hormone medication. The DM group and CG received tap drinking water ad libitum.

The sample size was estimated based on previous studies where 5–10 animal assays could demonstrate distinct functional and molecular characteristics in the LG of groups DM and HT compared to healthy male adult rats. The reason for different time lengths of experimental period for each group, after they complete eight weeks of life, is based on observations from previously published works and pilot studies indicating the minimum period to develop ocular manifestations (i.e., 8 weeks for DM and 5 weeks for HT) [9,11]. More extended periods of hormone deprivation induce excessive systemic impairment and impact several organic functions [15].

### 4.2. Behavioral Studies

To investigate whether DE is caused by hormonal deprivation, an eye wipe test was performed. After setting the animals, the room, and the acrylic chambers for an hour, the right eyes of all groups (*n* = 10 animals/group) were treated with the instillation of 20 µL of capsaicin at 10 μM diluted in PBS at pH 7.2 and 25 °C (Sigma Aldrich Brasil Ltda, Cotia, SP, Brazil). A drop of capsaicin was instilled with each eye opened by hand, using a micropipette. The behavioral response to pain was evaluated based on the number of times the right paw wiped the eye, which was recorded with a digital camera (DSC-W5, Sony, Tokyo, Japan) for 3 min after capsaicin installation. Eye wipe movements were counted from digital films by a masked observer using an iMac computer and compared between groups (Apple Inc., Cupertino, CA, USA).

### 4.3. Clinical Evaluation

The functional impact of DM and HT on the TF and OS was evaluated as previously described [26]. At the determined endpoint (i.e., eight weeks for DM group, five weeks for HT group, and six weeks for CG), the animals were evaluated after anesthesia with an intraperitoneal injection of ketamine (5 mg/100 g b.w.) (União Química Farmacêutica S.A, Embu-Guaçu, SP, Brazil) and xylazine (2 mg/100 g b.w.) (Laboratorio Callier S.A., Barcelona, Spain) to collect the following data:Tear film Osmolarity: tear samples were harvested from the animals without stimulation or need for eye drops, with the approximation of a delicate collector, and the measurements were made using Osmometer Tearlab^®^ osmolarity system (San Diego, CA, USA).Corneal fluorescein staining: an eye drop of 1% sodium fluorescein (Allergan Produtos farmacêuticos LTDA, Guarulhos, Sao Paulo, Brazil) was applied in the right eye to identify the areas and categorize the intensity of the keratitis, with grades from 0 to 15.Phenol red thread test was applied to measure the tear flow (TF) in millimeters for 30 s, as previously described (Showa Yakuhin Kako Co., Ltd., Tokyo, Japan and Menicon USA Inc., Clovis, CA, USA) [35].Blood Osmolarity was determined by measuring the freezing point of the blood plasma solution (Advanced instruments—Two Technology Way, Norwood, MA, USA).

### 4.4. Histological Evaluation

The effects of DM and HT on the LFU structures were also investigated using histology. After euthanasia with an excess of thiopental sodic (1000 mg/kg; Laboratório Cristalia, Sao Paulo, SP, Brazil), LG and CO tissues were harvested. The LG was collected, fixed in 17% formaldehyde (Merck KGaA, Darmstadt, Germany) for 24 h, dehydrated in an increasing series of ethanol (Merck KGaA, Darmstadt, Germany), diaphanized in xylol (Merck KGaA, Darmstadt, Germany), and soaked in paraffin (Merck KGaA, Darmstadt, Germany) baths at 60 °C for inclusion in blocks.

The six μm thick slices were cut on the Leica Jung RM2065 microtome (Leica Microsystems GmbH, Wetzlar, Germany). The sections were dewaxed in xylol, rehydrated in a decreasing ethanol series, and washed in distilled water. They were then stained with hematoxylin solution for 2 min, washed in running water, immersed in eosin for 2 min, and washed again in running water. Subsequently, the slides were dehydrated in increasing ethanol series, diaphanized in xylol, and mounted on tissue mount (Tissue Tek^®^ Glas ™ Mounting Media, Sakura, Finetek, Nagano, Japan). They were compared with CG. The slides of CO and LG were compared using descriptive analysis. The mean thickness of central cornea epithelia was compared among the samples of the three groups (*n* = 10 animals per group).

### 4.5. Anti-Oxidant Enzymes and Oxidative Stress Products

To investigate its role and repercussions of oxidative stress in the LFU, we compared oxidative enzyme activity in the LG of the DM, HT, and control groups at their endpoints mentioned above.

*Glutathione disulfide* (*oxidized glutathione*): LGs from the three groups (*n* = 10/group) were homogenized in a standard buffer provided by the producer of the analysis Glutathione Fluorometric Assay Kit (BioVision Incorporated, Milpitas, CA, USA). Each sample was normalized to provide 40 mg of LG tissue. The reactions were processed as indicated in the producer protocol. The sample and control fluorescence was read at the following excitation/emission wavelength 340/420 nm using a spectrophotometer (SpectroMax M3, Molecular Devices, Sunnyvale, CA, USA). 

*Catalase*: again, LGs of DM, HT, and control groups (*n* = 10/group) were homogenized, and 100 mg of tissue sample was submitted to the analysis according to the producer protocol (Catalase Assay Kit, BioVision Incorporated, Milpitas, CA, USA). The colorimetric measurement was performed at the wavelength of 570 nm in a plate reader using the spectrophotometer, as mentioned above. The concentrations of H_2_O_2_ were obtained and compared to a standard curve provided by the manufacturer and among the groups.

### 4.6. Quantitative Real-Time PCR

The rats were euthanized using ketamine (União Química Farmacêutica S.A, Embu-Guaçu, SP, Brazil) (5 mg/100 g body weight), xylazine (Laboratorio Callier S.A., Barcelona, Spain) (2 mg/100 g b.w.), and thiopental sodic (Laboratório Cristália, São Paulo, SP, Brazil) (1000 mg/kg). The right CO, LG, and TG tissues from the rats of the 3 groups were harvested and embedded in RNAlater Solution (Ambion, Waltham, MA, USA) and stored at −80 °C until we proceeded to RNA extraction protocol for real-time quantitative PCR (qPCR) analysis. The relative expression of the mRNA of pro-inflammatory cytokines *Il1b*, *Il6*, *Tnfa*, and *Mmp9* was compared in LG, CO, and TG samples from control, DM, and HT groups. In addition, the relative expression of the mRNA of the tissue repair elements in LG *Bmp7*, *Runx1*, *Runx3*, *Fgf10*, and *Smad1* was compared to those groups.

qPCR used hydrolysis probes (Applied Biosystems, Carlsbad, CA, USA). The RNA samples were extracted using RNeasy Mini Kit (Qiagen, Germantown, MD, USA) and quantified using a spectrophotometer NanoDrop 2000 c (Thermo Scientific, Wilmington, DE, USA).

Samples containing 500 ng of total RNA of CO tissue, 1000 ng of total RNA of LG tissue, and 150 ng of total RNA of TG tissue were used to produce the cDNA with the QuantiTect Reverse Transcription Kit (Qiagen, Germantown, MD, USA) in the ProFlex PCR System (Applied Biosystems, Carlsbad, CA, USA).

qPCR was conducted using ViiA7 Real-time PCR System (Applied Biosystems, Carlsbad, CA, USA). The hydrolysis probes in this study were the following: Rn.PT 5838028824 (*Il1b*), Rn.PT 5813840513 (*Il6*), Rn.PT 5811142874 (*Tnf*), Rn.PT 587383134 (*Mmp9*), Rn.PT 5810180444 (*Bmp7*), Rn.PT 5810814634 (*Fgf10*), Rn.PT 589220704.g (*β-actin*) (all these from IDT); Rn00565555_m1 (*Smad1*), Rn00569082_m1 (*Runx1*), Rn00590466_m1 (*Runx3*) (Applied Biosystems, Carlsbad, CA, USA). The amplification was performed in duplicate with 5.5 uL of QuantiNova Probe PCR Kit (Qiagen, Germantown, MD, USA), 0.5 μL of hydrolysis probe, and 4.5 μL of 1:4 dilution of the cDNA in a total volume of 10 μL. The cycles for qPCR were as follows: one cycle of 95 °C for 2 min, 50 cycles of 5 s at 95 °C, and 19 s at 60 °C.

The Thermo Fisher Cloud Software, RQ version 3.7, determined the relative quantification (Life Technologies Corporation, Carlsbad, CA, USA). The results were compared in each tissue sample of three groups.

### 4.7. Statistical Analysis

Prism 8.0 (GraphPad Software, San Diego, CA, USA) was used for descriptive statistics and to compare the response of DM and HT groups with CG using one-tailed, Mann–Whitney U statistical tests. The level of significance was set at *p* < 0.05.

## 5. Conclusions

In conclusion, our study shows that HT and DM induce changes in the LFU, compatible with the DE mechanism, but not limited to the CO or LG. It also confirms the difficulties presented in the clinical studies intended to characterize the standards of DE changes induced by chronic hormone disturbance where other influences, such as disparity in genetics, disease time course, and treatment adherence, and environmental influences in the natural history of the disease take place and create more heterogeneity in the manifestations of DE. We hope the present observations will contribute to future studies on more specific and sensitive diagnostic tools and customized therapeutic approaches for DE.

## Figures and Tables

**Figure 1 ijms-24-06974-f001:**
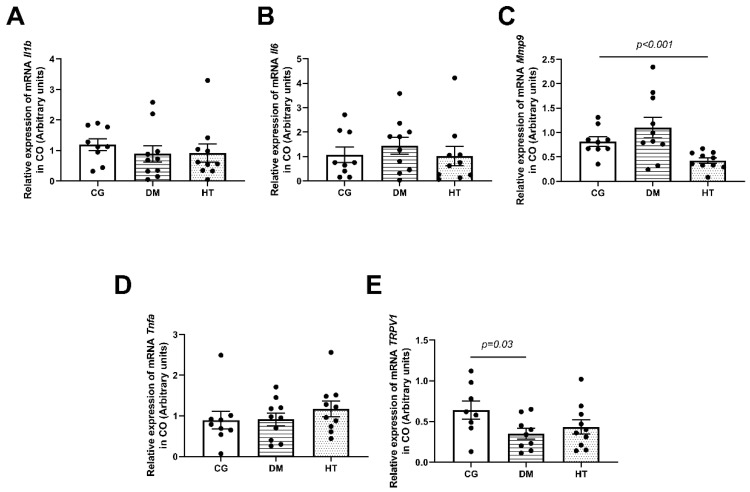
The relative mRNA expression of pro-inflammatory cytokines in CO of the DM, HT, and control groups. The data revealed lower expression of *Mmp9* mRNA in HT group (*p* < 0.001) (**C**), and no changes in the *Il1b* (**A**), *Il6* (**B**), and *Tnf* (**D**) compared to the CG. (**E**) *TRPV1* relative mRNA expression in the CO. The data revealed a lower expression of *TRPV1* mRNA in the DM group (*p* = 0.03). In all tests, *n* = 10/animals per group. The comparisons were made between DM or HT groups and CG using the Mann–Whitney U test. Each animal value is represented by a black dot in the figure.

**Figure 2 ijms-24-06974-f002:**
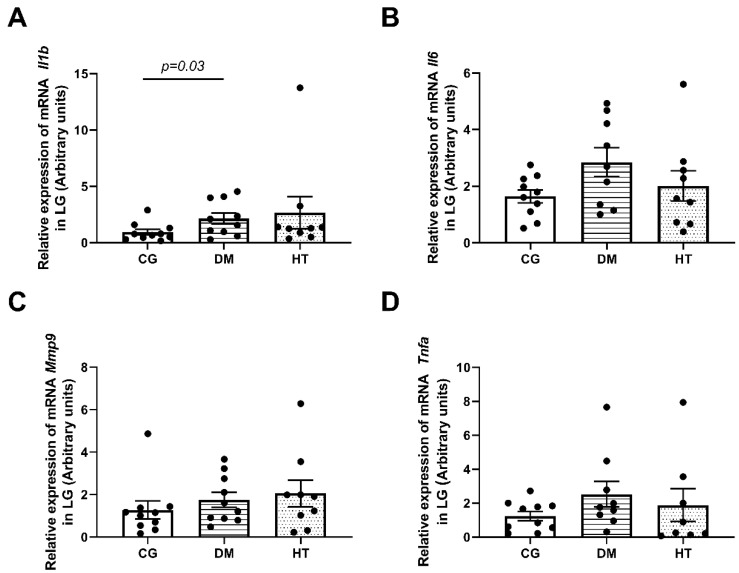
Pro-inflammatory cytokines relative mRNA in the LG. DM group shows a higher expression of *Il1b* mRNA (*p* = 0.03) (**A**). The other cytokines did not present a significant difference from CG (**B**–**D**). In all tests, *n* = 10/animals per group. The comparisons were made between DM or HT groups and CG using the Mann–Whitney U test. Each animal value is represented by a black dot in the figure.

**Figure 3 ijms-24-06974-f003:**
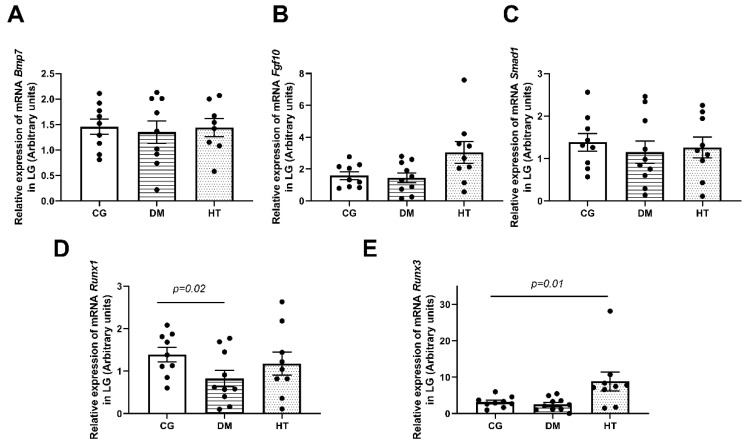
Relative mRNA expression of pro-mitotic elements in the LG of the DM, HT, and control groups at their final experimental period (**A**–**E**). The results show a lower expression of *Runx1* mRNA in DM group (*p* = 0.02) (**D**), and a higher expression of *Runx3* in HT group (*p* = 0.01) compared to the CG (**E**). No changes were observed in other pro-mitotic elements (**A**–**C**). The *n* = 10/group and the comparisons between DM or HT groups and the CG were made using the Mann–Whitney U test. Each animal value is represented by a black dot in the figure.

**Figure 4 ijms-24-06974-f004:**
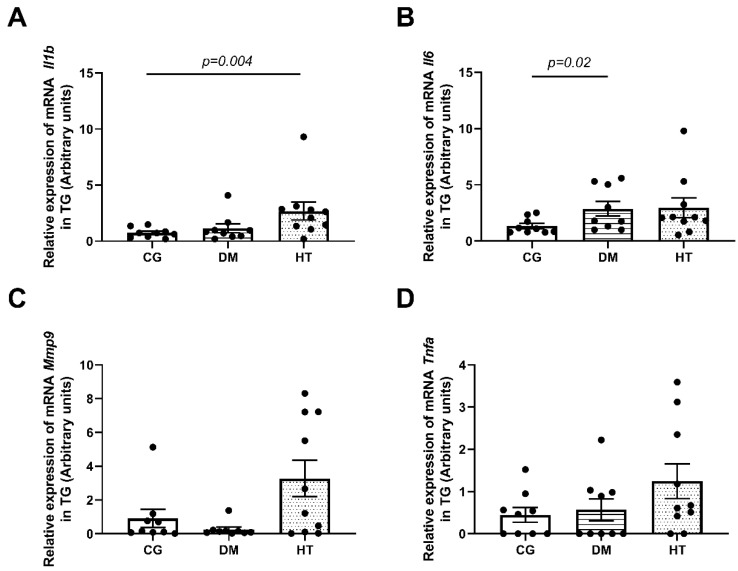
Relative mRNA expression of pro-inflammatory cytokines in the TG. The data revealed a higher expression of *Il1b* mRNA in HT group (*p* = 0.004) (**A**) and a higher expression of *Il6* mRNA in DM group (*p* = 0.02) (**B**). The expression of *Mmp9* and *Tnfa* mRNA was not different in the groups (**C**,**D**). In all tests, *n* = 10/animals per group. The comparisons were made using the Mann–Whitney U test. Each animal value is represented by a black dot in the figure.

**Table 1 ijms-24-06974-t001:** Comparative functional and laboratory paired analysis between DM or HT groups and the control group. The comparisons were made between experimental groups and the control group at their final experimental period using the Mann–Whitney U test. In all tests, *n* = 10/animals per group. Significant values were written in bold.

Experimental Groups	Control	DM	HT	* Values of *p*
**Eye wipe test (movement/s)**	4.2 ± 3.3	6.4 ± 2.7	4.0 ± 3.0	***p* > 0.05**
**Phenol red thread (mm/30 s)**	8.3 ± 4.1	**4.7 ± 3.5 ***	6.4 ± 4.4	***p* = 0.02**
**Tear osmolarity (mOsm/L)**	298.4 ± 17.3	321.8 ± 33.5	**338.1 ± 13.5 ***	***p* < 0.001**
**Blood osmolarity (mOsm/L)**	302.2 ± 5.8	**341.2 ± 16.4 ***	321.5 ± 27.4	***p* < 0.001**
**Epithelial thickness of cornea (μm)**	26.1 ± 1.9	**27.3 ± 2.4 ***	24.7 ± 2.6	***p* = 0.003**
**Catalase activity (mU/mL)**	63.8 ± 2.5	**68.7 ± 2.0 ***	**66.3 ± 1.9 ****	*** *p* < 0.001** **** *p* = 0.02**

* *p*-value on comparison between DM and control group and ** *p*-value on comparison between the control group and HT.

## Data Availability

Links to publicly archived datasets analyzed or generated during the study: https://uspdigital.usp.br/repositorio/solicitar-repositorio.jsp?codmnu=9975, accessed on 30 January 2023.

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
