# Peer review of "Distinct Inflammatory and Oxidative Effects of Diabetes Mellitus and Hypothyroidism in the Lacrimal Functional Unit"

_ijms, 2023, doi:10.3390/ijms24086974_

Round 1
Reviewer 1 Report
In this study, the authors investigate the inflammatory and oxidative effects of diabetes mellitus (DM) and hypothyroidism (HT) on the lacrimal functional unit in the context of dry eye. Both functional and molecular assays have been conducted. The topic is interesting and meaningful. To further improve the manuscript, the authors are invited to address the following questions.
My comments:
1. There are many tests for the diagnosis of dry eye. Is it sufficient to confirm the establishment of dry eye in the rat model with tear flow or osmolarity as the diagnostic criteria? (The authors claim that “the tear flow showed large variability in the group, confirming the low predictive value of the PRT test” and that “the present data confirm the difficulty in characterizing the DE with a single test”.)
2. MMP9 has been used as a diagnostic maker of dry eye (commercial products for this purpose are available, e.g., RPS InflammaDry from Rapid Pathogen Screening Inc.). The authors are invited to discuss why MMP9 showed lower expression in the HT group. As it is not common for dry eye.
3. I have not found the data that support the claim “The present data confirm the difficulty in characterizing the DE with a single test”. The authors are invited to discuss this point in more detail.
4. Please explain why non-parametric tests were used for statistical analysis.
5. The authors claim that “the present work identified the mechanisms and the targets of the effects of insulin and thyroid hormone deprivation to the LFU”. However, in my opinion, the data presented in the manuscript are not enough to identify the mechanisms. Therefore, the conclusion should be revised.
6. The authors present the protocol for corneal fluorescein staining in the “Material and Methods” section. However, no results have been presented in the manuscript.
7. Line 32-33: The Ocular Surface is a concept which includes the cornea and conjunctiva.
8. Line 58-59: The OS in the sentence “this study aims to evaluate the functional and molecular inflammatory aspects of the LG, OS, and TG in the following two rat models of DE” is suggested to revised as CO (cornea), as the authors haven’t investigated the other components of the ocular surface.
9. The caption of table 1 is too redundant, and should be revised to be concise. Many contents in the caption have already been presented in the main text.
10. In the manuscript, Dye Eye is sometimes abbreviated as DE, sometimes abbreviated as DES (e.g., the functional manifestations of DE in the HT and DM revealed distinct clinical profiles, confirming the large spectrum of DES manifestations). This is also the case for abbreviations of the control group (sometimes abbreviated as CG, e.g., figures 2, 4; sometimes abbreviated as CT, e.g., figures 1, 3, 5;). The authors are asked to keep consistent.
11. The statement “The comparisons were made between DM or HT and CG group at their final experimental period (five weeks for HT and eight weeks for DM).” appears several times in the manuscript. I suggest the authors only indicate this information once in the “Material and Methods” section.
12. The authors use too many abbreviations in the text, which compromises the readability of the manuscript. Especially considering that some of the abbreviations are not defined at the first mention (e.g., CG).
13. It is strange that a “conclusion” section is presented following “Material and Methods”.
14. Please check the grammar and spelling. For example:
l Line 143: mRNa changes; Line 173: a ke factor; Line 183: Already in DM; Line 185: stecuture
l Line 190-191: We also observed that several aspects of the DM and HT diseases are distinct among DM and HT diseases
l Line 264-265: Blood Osmolarity, precisely measured the freezing point of the solution of blood plasma, the osmolality was determined
Author Response
Reply to reviewer 1:
We would like to thank the reviewer #1 for the relevant comments and suggestions on our manuscript ijms-2222418, entitled "Distinct inflammatory and oxidative effects of diabetes mellitus and hypothyroidism in the lacrimal functional unit". All suggestions were taken into consideration, and modifications have been made accordingly.
Reviewer #1 Comments:
In this study, the authors investigate the inflammatory and oxidative effects of diabetes mellitus (DM) and hypothyroidism (HT) on the lacrimal functional unit in the context of dry eye. Both functional and molecular assays have been conducted. The topic is interesting and meaningful. To further improve the manuscript, the authors are invited to address the following questions.
We would like to express our grattitude to Reviewer # 1 for these encouraging comments on our work.
- There are many tests for the diagnosis of dry eye. Is it sufficient to confirm the establishment of dry eye in the rat model with tear flow or osmolarity as the diagnostic criteria? (The authors claim that “the tear flow showed large variability in the group, confirming the low predictive value of the PRT test” and that “the present data confirm the difficulty in characterizing the DE with a single test”.)
The diagnosis of dry eye is a challenge for human beings and other species. A study conducted by McCarty CA et al. with more than 900 individuals in Melbourne, Australia, and published in the Ophthalmology journal in 1998, revealed that prevalence could change from 1.5% to 16.6 % considering pre-established test thresholds. In that study, only one individual was positive for dry eye in all the tests. The higher resistance and lack of tools to investigate rodent symptoms make the dry eye diagnosis more challenging. Our studies use several "in vivo" tools and consider the positivity in at least one sufficient to indicate dry eye. Corneal fluorescein staining, tear flow and tear film osmolarity were used in the present work. Corneal fluorescein staining and corneal cells impression cytology are much less sensitive tests, negative even in moderate cases of dry eye in animal models.
- MMP9 has been used as a diagnostic maker of dry eye (commercial products for this purpose are available, e.g., RPS InflammaDry from Rapid Pathogen Screening Inc.). The authors are invited to discuss why MMP9 showed lower expression in the HT group. As it is not common for dry eye.
MMP9 in the tear film is considered a valuable qualitative test to identify dry eye in people with 85% of sensitivity and 94% of specificiity compared to health controls (Sambrusky R et al, 2013). The limitaiton is that as an innate inflammatory marker it reponds to various other inflammatory challenges to the OS. The signnficant lower levels of MMP9 mRNA in corneas of the HT group compared to controls suggest that inflammation is not a key early event in the cornea as it is relevant in the TG of the same group, where IL1-ß and TNF-alfa mRNA levels are significantly higher. To address this point we added the following phrase to the Discussion section (page 08, para 02):
The signnficant lower levels of MMP9 mRNA in corneas of the HT group, and lack of changes of other cytokines compared to controls suggest that inflammation is not a key early event in the cornea as it is relevant in the TG of the same group, where IL1-ß mRNA levels are significantly higher.
- I have not found the data that support the claim “The present data confirm the difficulty in characterizing the DE with a single test”. The authors are invited to discuss this point in more detail.
The reviewer is right. The statement is confusing. The whole paragraph was rephrased as follows (page 08, para 06):
The present data shows no difference in the eye wipe test between DM, HT and CG, and a high standard deviation on tear flow values (phenol red thread) in the HT group and the tear film osmolarity results in the DM group. It reveals that clinical or functional tests are limited in characterizing DE once applied as a single test. This variability of results in distinct groups, even in experimental studies where we can homogenize the genetic background, the time course to measure the parameters in the initial phase of the DE disease, and the environmental conditions is challenging in studies addressing DE mechanisms
- Please explain why non-parametric tests were used for statistical analysis.
The normal distribution of the continuous data was investigated by the Shapiro-Wilk test. Since the data deviated from a normal distribution, the comparisons between the groups HT, DM and control were made using the Mann-Whitney U test.
- The authors claim that “the present work identified the mechanisms and the targets of the effects of insulin and thyroid hormone deprivation to the LFU”. However, in my opinion, the data presented in the manuscript are not enough to identify the mechanisms. Therefore, the conclusion should be revised.
We agree with the Reviewer # 1 and rephrased the conclusion, enphasizing we identified elements of the mechanisms as follows (Page 08, para 05):
In summary, the present work identified elements of the inflammatory, anti-oxidant, and neural pathways on the mechanisms and the targets of the effects of insulin and thyroid hormone deprivation on the LFU, (...).
- The authors present the protocol for corneal fluorescein staining in the “Material and Methods” section. However, no results have been presented in the manuscript.
The reviewer is correct. The outcome of corneal staining was inadvertently suppressed, Corneal fluorescein staining was performed, and no changes were observed in the groups. The following phrase was added to the results to clarify it (Page 02, para 07):
Corneal fluorescein staining did not show differences between the DM, HT, and CG.
- Line 32-33: The Ocular Surface is a concept which includes the cornea and conjunctiva.
The reviewer is correct. The statement was rephrased for accuracy (page 02, para 02):
The Lacrimal Functional Unit (LFU) system includes the LG, the cornea, the conjunctiva (therefore the Ocular Surface), the Meibomian glands, the eyelids, and the sensory nerves.
- Line 58-59: The OS in the sentence “this study aims to evaluate the functional and molecular inflammatory aspects of the LG, OS, and TG in the following two rat models of DE” is suggested to revised as CO (cornea), as the authors haven’t investigated the other components of the ocular surface.
We agree with the Reviewer # 1 recommendation and replaced OS by CO, as follows (page 02 para 06):
Therefore, this study aims to evaluate the functional and molecular inflammatory aspects of the LG, CO, and TG in the following two rat models of DE: DM and HT.
- The caption of table 1 is too redundant, and should be revised to be concise. Many contents in the caption have already been presented in the main text.
The legend of table 1 was rewritten for concision. The p values were removed and limited to their columns, and the units of measurement were added to their specific rows (highlighted in yellow). The proposed legend is presented below (page 03):
Table 1. Comparative functional and laboratory paired analysis between DM or HT and Control. The comparisons were made between experimental groups from the Control group at their final experimental period (five weeks for HT and eight weeks for DM) with the Mann-Whitney U test. In all tests, n=10/animals per group.
- In the manuscript, Dye Eye is sometimes abbreviated as DE, sometimes abbreviated as DES (e.g., the functional manifestations of DE in the HT and DM revealed distinct clinical profiles, confirming the large spectrum of DES manifestations). This is also the case for abbreviations of the control group (sometimes abbreviated as CG, e.g., figures 2, 4; sometimes abbreviated as CT, e.g., figures 1, 3, 5;). The authors are asked to keep consistent.
Thank you for the observation and suggestions. a) DE was the abbreviation chosen and replaced in the whole manuscript. b) the control group abbreviation CG was adopted in the text and figures. The Figure 2 was merged to Figure 1and the figures renumbered in the whole manuscript relplacing CT by CG.
- The statement “The comparisons were made between DM or HT and CG group at their final experimental period (five weeks for HT and eight weeks for DM).” appears several times in the manuscript. I suggest the authors only indicate this information once in the “Material and Methods” section.
The statement was limited to "Material and Methods" section and removed from other parts of the manuscript.
- The authors use too many abbreviations in the text, which compromises the readability of the manuscript. Especially considering that some of the abbreviations are not defined at the first mention (e.g., CG).
Thank you for the suggestion. The excess of abbreviations was removed from the manuscript (ex: CNS). The abbreviations of the groups (CG, HT, and DM) and organs evaluated (CO, LG, and TG) were maintained to facilitate the correlation between the text and figure.
- It is strange that a “conclusion” section is presented following “Material and Methods”.
The sequence of sections was indicated in the IJMS journal Author's guidelines. We bold-highlighted the Conclusions section to highlight this part of the manuscript.
- Please check the grammar and spelling. For example:
l Line 143: mRNa changes; Line 173: a ke factor; Line 183: Already in DM; Line 185: stecuture
l Line 190-191: We also observed that several aspects of the DM and HT diseases are distinct among DM and HT diseases
l Line 264-265: Blood Osmolarity, precisely measured the freezing point of the solution of blood plasma, the osmolality was determined
The text was revised, and those grammar and spelling errors were corrected.
Please, see the complete authors' reply attached.

Reviewer 2 Report
This is an interesting article presenting data on analysis of lacrimal functional unit in rat models of diabetes and hypothyroidism; minor revisions are suggested before possible publication.
-The main limitations of this study should be reported in the Discussion section.
-The fact that the three groups had 10 animals per group (line 220) should be better reported.
-Meaning of CG and EWT (line 63), GSSG (line 86), should be provided.
-The Conclusions section should represent a distinct paragraph.
-A thorough English revision should be performed (lines 172-173, 237, 264-265…).
Author Response
Reply to reviewer # 2:
We would like to thank the reviewer #1 for the relevant comments and suggestions on our manuscript ijms-2222418, entitled "Distinct inflammatory and oxidative effects of diabetes mellitus and hypothyroidism in the lacrimal functional unit". All suggestions were taken into consideration, and modifications have been made accordingly.
Reviewer #2 Comments:
This is an interesting article presenting data on analysis of lacrimal functional unit in rat models of diabetes and hypothyroidism; minor revisions are suggested before possible publication.
We would like to thank to Reviewer # 1 for these encouraging comments on our work.
- The main limitations of this study should be reported in the Discussion section.
The following paragraph was added to the Discussion section to address the major limitations of the study (Page 09, para 01):
This study's weaknesses are the lack of long-term DE and the response to DM and HT therapy. Moreover, considering the higher prevalence of DE in the female sex in several species, the present analysis with male rats is limited in identifying the mechanisms of DE in both sexes.
- The fact that the three groups had 10 animals per group (line 220) should be better reported.
We described our criteria to determine this sample size and included them in the Methods section as follows (Page 09, para 07):
The sample size was estimated based on previous studies where 5-10 animal assays could demonstrate distinct functional and molecular characteristics in the LG of groups DM e HT compared to healthy male adult rats.
- Meaning of CG and EWT (line 63), GSSG (line 86), should be provided.
We clarifed those abbreviations in the first tie they appeared in the text, as indicated by yellow mark.
- The Conclusions section should represent a distinct paragraph.
The conclusion section was separated from the Methdos section using a bold subtitle in a distinct section (5).
- A thorough English revision should be performed (lines 172-173, 237, 264-265…).
English review was applied to the whole text, including the indicated sentences.
Please, see the complete authors' reply attached.

Round 2
Reviewer 1 Report
My comments are addressed adequately and I have no further questions.